# Size-dependent patterns of cell proliferation and migration in freely-expanding epithelia

Matthew A Heinrich[1], Ricard Alert[2,3], Julienne M LaChance[1], Tom J Zajdel[1], Andrej Košmrlj[1,4], Daniel J Cohen[1]*

[1]Department of Mechanical and Aerospace Engineering, Princeton University, Princeton, United States; [2]Lewis-Sigler Institute for Integrative Genomics, Princeton University, Princeton, United States; [3]Princeton Center for Theoretical Science, Princeton University, Princeton, United States; [4]Princeton Institute for the Science and Technology of Materials (PRISM), Princeton University, Princeton, United States

**Abstract** The coordination of cell proliferation and migration in growing tissues is crucial in development and regeneration but remains poorly understood. Here, we find that, while expanding with an edge speed independent of initial conditions, millimeter-scale epithelial monolayers exhibit internal patterns of proliferation and migration that depend not on the current but on the initial tissue size, indicating memory effects. Specifically, the core of large tissues becomes very dense, almost quiescent, and ceases cell-cycle progression. In contrast, initially-smaller tissues develop a local minimum of cell density and a tissue-spanning vortex. To explain vortex formation, we propose an active polar fluid model with a feedback between cell polarization and tissue flow. Taken together, our findings suggest that expanding epithelia decouple their internal and edge regions, which enables robust expansion dynamics despite the presence of size- and history-dependent patterns in the tissue interior.

*For correspondence:
danielcohen@princeton.edu

Competing interests: The authors declare that no competing interests exist.

## Introduction

Writing in 1859, physiologist Rudolf Virchow presented the concept of the 'Zellenstaat' or 'Cell State,' describing tissues as 'a society of cells, a tiny well-ordered state' (*Virchow, 1855*). This social framework motivated *Abercrombie and Heaysman, 1954* work on cellular behavior that elucidated how encounters between cells can regulate locomotion and proliferation via contact inhibition. Since then, concerted interdisciplinary effort has been brought to bear on understanding how cell-cell interactions give rise to the complex collective behaviors driving so many crucial biological processes. One of the most foundational collective behaviors is collective cell migration—the directed, coordinated motion of cellular ensembles that enables phenomena such as gastrulation, wound healing, and tumor invasion (*Friedl and Gilmour, 2009*). Given this importance, considerable effort spanning biology, engineering, and physics has been directed towards understanding how local cellular interactions can give rise to globally coordinated motions (*Alert and Trepat, 2020*; *Hakim and Silberzan, 2017*).

Studies of collective cell migration are most often performed using epithelial tissues due to their fundamental role in multicellular organisms and strong cell-cell adhesion, which in turn gives rise to elegant, cohesive motion. Moreover, given that epithelia naturally form surfaces in vivo, studying epithelial layers in vitro has a physiological basis that can inform our understanding of processes such as healing (*Poujade et al., 2007*), envelopment (*Steinberg, 2007*), and boundary formation (*Dahmann et al., 2011*). These features have made epithelia both the gold standard in collective cell migration studies, and one of the most well-studied models for biological collective behaviors.

**eLife digest** Cells do not exist in isolation. Instead, they form tissues, where individual cells make contact with their neighbors and form microscopic 'architectures'. Epithelia are a type of tissue where cells are arranged in flat sheets, and are found in organs such as the lining of the kidney or the skin.

Tissues need to grow, especially early in life. If tissues are damaged – for example, if the skin is cut or grazed – cells also need to divide (to create new healthy cells) and move as a group (to close the wound). Such coordinated motions result in cells exhibiting distinct group behaviors, similar to those observed within crowds of people or schools of fish. If coordination breaks down, problems can happen such as uncoordinated tissue growth seen in cancer.

However, how cell movements are coordinated is still not fully understand. For example, researchers know that cells' positions within a group can determine how they behave, meaning that even the same type of cell could behave differently at the edge or center of a tissue. This suggests that the initial size and shape of a tissue should influence its subsequent growth and behavior; however, the nature of this influence is still largely unknown. Heinrich et al. therefore wanted to determine the differences in the way larger and smaller tissues grow.

Microscope imaging was used to track the growth of circular, artificial tissues made from single-layered sheets of dog kidney cells grown in the laboratory. Comparing how quickly the tissues expanded revealed that the area of tissue circles that started out smaller increased at a much faster rate than that of tissue circles that were larger to begin with. This turned out to be because the edges of the tissues grew at a constant speed, independent of their initial size or shape, but circles with a smaller area have a larger proportion of cells on their edges. The motions of the cells at the center of the tissues had no effect on how the edges of the tissue grew. A final observation was that the way tissues of a given size behaved depended on whether they had grown to be that size, or they started off that big.

These results shed light on how groups of cells interact in growing tissues. In the future, this information could be used to predict how different tissues grow over time, potentially helping scientists engineer better artificial tissues or organs for transplantation.

Due to the complexity of collective behaviors, much effort has gone towards reductionist assays that restrict degrees of freedom and ensemble size to simplify analysis and interpretation. One such approach is to confine a tissue within predefined boundaries using micropatterning to create adhesive and non-adhesive regions (*Doxzen et al., 2013*; *Deforet et al., 2014*; *Notbohm et al., 2016*; *Pérez-González et al., 2019*; *Peyret et al., 2019*; *Petrolli et al., 2019*). Such confinement mimics certain in vivo contexts such as constrained tumors as well as aspects of compartmentalization during morphogenesis (*Lecuit and Lenne, 2007*). Alternately, many studies have explored the expansion of tissues that initially grow into confluence within confinement but are later allowed to migrate into free space upon removal of a barrier. A popular assay of this type relies on rectangular strips of tissue that are allowed to expand in one or both directions (*Poujade et al., 2007*; *Trepat et al., 2009*; *Petitjean et al., 2010*; *Reffay et al., 2011*; *Nnetu et al., 2012*; *Serra-Picamal et al., 2012*; *Zhang et al., 2017*; *Uroz et al., 2018*; *Tlili et al., 2018*), where averaging along the length of the strip can reveal coordinated population-level behaviors such as complex migration patterns, non-uniform traction force fields, and traveling mechanical waves. Other studies have focused on the isotropic expansion of micro-scale (< 500 μm diameter) circular tissues using the barrier stencil technique (*Jang et al., 2017*) as well as photoswitchable substrates (*Rolli et al., 2012*). Still more work has explored approaches to induce directional migration, from geometric cues to applied electric fields (*Vedula et al., 2012*; *Cohen et al., 2014*).

In contrast to micro-scale confinement assays, other work has focused on large, freely-expanding tissues of uncontrolled initial size and shape, which grow from either single cells (*Puliafito et al., 2012*; *Huergo et al., 2011*) or cell-containing droplets (*Lee et al., 2013*; *Beaune et al., 2014*). Related experiments track long-term growth of cell colonies via images taken once per day over several days, but this low temporal resolution cannot access timescales over which migration is important (*Huergo et al., 2011*; *Simpson et al., 2013*). Thus, there is still a lack of assays to study long-

term expansion and growth of large-scale tissues with precisely-controlled initial conditions, especially initial tissue size, shape, and density.

To address this gap, we leveraged bench-top tissue patterning (*Poujade et al., 2007*; *Cohen et al., 2016*) to precisely pattern macro-scale circular epithelia of two sizes (>1 mm in diameter) and performed long-term, high frequency, time-lapse imaging after release of a barrier. To elucidate the consequences of size effects on the tissue, we tracked every cell, relating the overall expansion kinetics to cell migration speed, cell density, and cell-cycle dynamics. We find that, whereas the tissue edge dynamics is independent of the initial conditions, the tissue bulk exhibits size-dependent patterns of cell proliferation and migration, including large-scale vortices accompanied by dynamic density profiles. Together, these data comprise the first comprehensive study of macro-scale, long-term epithelial expansion, and our findings demonstrate the importance of exploring collective cell migration across a wider range of contexts, scales, and constraints.

## Results

### Expansion of millimeter-scale epithelia of different sizes and shapes

We began by characterizing the overall expansion and growth of tissues with the same cell density but different initial diameters of 1.7 mm and 3.4 mm (a 4X difference in area, with tissues hereafter referred to as either 'small' or 'large'), using an MDCK cell line stably expressing the 2-color FUCCI cell-cycle marker (*Sakaue-Sawano et al., 2008*; *Streichan et al., 2014*; *Uroz et al., 2018*; *Beaune et al., 2014*; *Benham-Pyle et al., 2016*). We patterned the tissues by culturing cells in small and large circular silicone stencils for ~18 hr (*Cohen et al., 2016*; *Poujade et al., 2007*), whereupon stencils were removed and tissues were allowed to freely expand for 46 hr (*Figure 1A*, *Figure 1— video 1*), while images were collected at 20 min intervals using automated microscopy (see Materials and Methods). Our cell seeding conditions and incubation period were deliberately tuned to ensure that the stencils did not induce contact inhibition of proliferation prior to stencil removal (checking FUCCI to ensure the tissue was not arrested in G1). Upon stencil removal, tissues expanded while maintaining their overall circular shape throughout the 2 day experiment. Unless otherwise noted, cell density at stencil removal was ~2700 cells/mm$^2$, a value consistent with active and growing confluent MDCK epithelia (*Streichan et al., 2014*; *Uroz et al., 2018*).

First, we measured relative areal increase (*Figure 1B*) and relative cell number increase (*Figure 1—figure supplement 1*) of small and large tissues. By 46 hr, small and large tissues had increased in area by 6.4X and 3.3X, respectively, while cell number increased by 9.2X and 5.5X, respectively. Since proliferation outpaces area expansion in long-term growth, average tissue density increased by the end of the experiment. The evolution of average tissue density was more complex, however, as small tissues experienced a density decrease from 4 to 12 hr while large tissues exhibited a monotonic increase in cell density (*Figure 1C*). Accordingly, at any given time after stencil removal, large tissues had a higher density than small tissues. Non-monotonic density evolution has been observed in thin epithelial strips (*Poujade et al., 2007*) and likely arises from competition between migration and proliferation dynamics, which we discuss later.

We then related area expansion to the kinematics of the tissue edge. To quantify edge motion, we calculated the average radial velocity of the tissue boundary, $v_r(t)$, at 1 hr intervals over 46 hr (Materials and methods). We found that $v_r$ is independent of both tissue size and a wide range of initial cell densities, in all cases reaching ~30 μm/h after ~16 hr (*Figure 1D*). Before reaching this constant edge velocity, $v_r$ ramps up during the first 8 hr after stencil removal, and, notably, overshoots its long-time value by almost 30%. We hypothesize that the overshoot is due to the formation of fast multicellular finger-like protrusions that emerge at the tissue edge in the early stages of expansion and then diminish (*Figure 1—video 2*). This hypothesis is supported by a recent model showing that edge acceleration (as observed during the first 8 hr in *Figure 1D*) leads to finger formation (*Alert et al., 2019*). It is remarkable that the edge radial velocity $v_r(t)$ is independent of the initial tissue size and density, especially considering that cell density evolution shows opposite trends at early stages of expansion for small and large tissues (*Figure 1C*). This observation suggests that the early stages of epithelial expansion are primarily driven by cell migration rather than proliferation or density-dependent decompression and cell spreading.

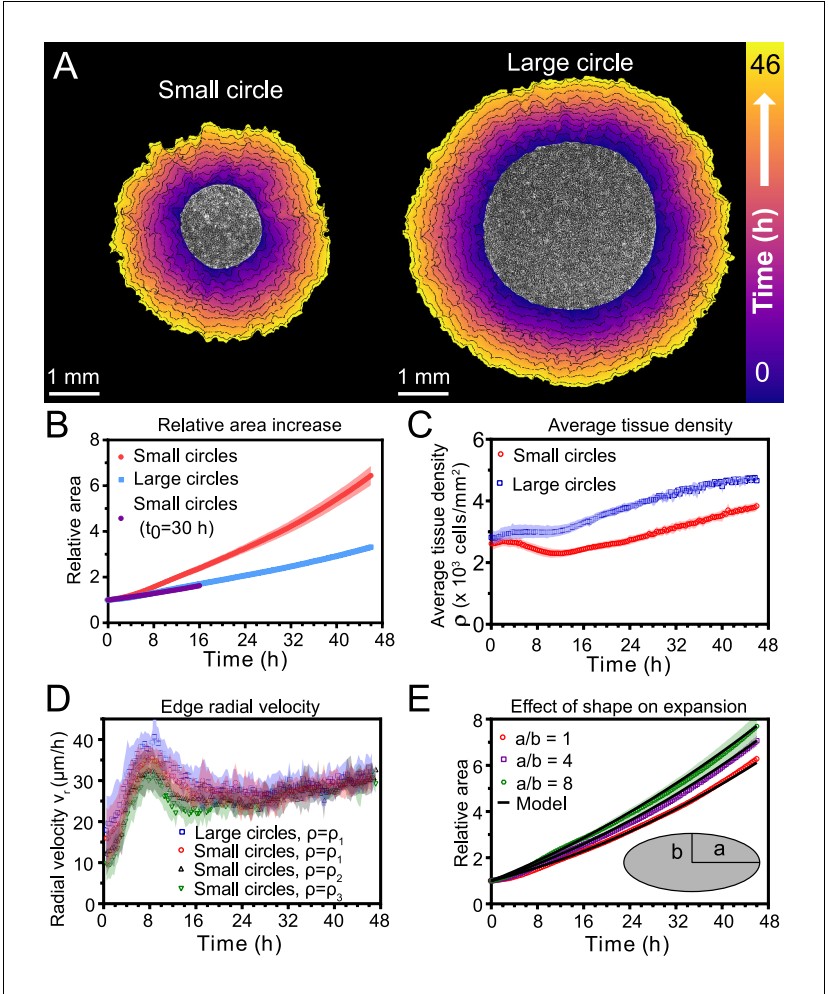

**Figure 1.** Expansion dynamics of millimeter-size cell monolayers. (**A**) Footprint throughout 46 hr growth period of representative small (left) and large (right) circular tissues, with the tissue outlines drawn at 4 h increments. Initial diameters were 1.7 mm and 3.4 mm. (**B**) Small circles exhibit faster relative area, $A(t)/A_0$, increase than large circles, where $A_0$ and $A(t)$ are the areas of tissues at the beginning of the experiment and at time $t$, respectively. Purple points show the relative area increase, $A(t + t_0)/A(t_0)$, of small tissues from the time $t_0 = 30$ h when they reached the size of the large circles. (**C**) Average tissue density $\rho(t) = N(t)/A(t)$ has non-monotonic evolution in small tissues but monotonically increases in large tissues, where $N(t)$ is the number of cells in a tissue at time $t$. (**D**) Edge radial velocity $v_r$ is largely independent of initial tissue size and cell density. We grouped initial cell densities as $\rho_1 = [2350, 3050]$ cells/mm$^2$, $\rho_2 = [1650, 2350]$ cells/mm$^2$, and $\rho_3 = [1300, 1650]$ cells/mm$^2$. (**E**) Experimental data on tissue shape and model fits. Assuming a constant migration speed $v_n$ in direction normal to the edge, we can predict the area expansion dynamics of elliptical tissues with different aspect ratios. The model fits our data for all tissues with $v_n \approx 29.5$ μm/hr, yielding normalized $\chi^2$ values of 0.79, 0.13, and 0.06 for aspect ratios of 8, 4, and 1 respectively ($\chi^2 < 1$ indicates a good fit; see Materials and methods). In B, data are from n = 16 tissues across five independent experiments (small and large circles). In C, n = 11 across four experiments for small circles, and n = 9 across three experiments for large circles. In D, n = 16 across five independent experiments for small and large circles, $\rho = \rho_1$; n = 13 across three experiments for small circles, $\rho = \rho_2$; and n = 11 across three experiments for small circles, $\rho = \rho_3$. In E, n = 4 across two experiments for a/b = 1 and a/b = 4, and n = 5 across two experiments for a/b = 8. Shaded regions correspond to standard deviations.

The online version of this article includes the following video and figure supplement(s) for figure 1:

**Figure supplement 1.** Relative proliferation in small and large tissues.

**Figure supplement 2.** Normal edge velocity $v_n$ of elliptical tissues at the major and minor axes.

**Figure 1—video 1.** Movie of expansion of representative small and large tissues in phase-contrast.
https://elifesciences.org/articles/58945#fig1video1

**Figure 1—video 2.** Finger-like protrusions emerge in the first 20 hr of tissue expansion.

*Figure 1 continued on next page*

*Figure 1 continued*

https://elifesciences.org/articles/58945#fig1video2

**Figure 1—video 3.** Expansion of sample elliptical cell monolayers with varying aspect ratios.
https://elifesciences.org/articles/58945#fig1video3

The observation that $v_r$ is independent of tissue size ought to explain why small tissues have faster relative area expansions than large tissues. We hypothesized that the relation between tissue size and areal increase could be attributed primarily to the perimeter-to-area ratio. Assuming a constant edge velocity $v_n$ normal to the tissue boundary, the tissue area increases as $dA = Pv_n dt$, where $P$ is the perimeter of tissue and $dt$ is a small time interval. Thus, the relative area increase $dA/A = (P/A)v_n dt$ scales as the perimeter-to-area ratio, which is inversely proportional to the radius for circular tissues, so the relative area increases faster for smaller tissues (*Figure 1B*).

To verify that the perimeter-to-area ratio is proportional to the relative area increase, we analyzed elliptical tissues with the same area and cell density but different perimeters (*Figure 1—video 3*). Increasing the perimeter-to-area ratio of a tissue by increasing its aspect ratio indeed leads to faster relative area expansion (*Figure 1E*). A simple, edge-driven expansion model with linear increase of the tissue major and minor axes predicts $A(t)/A(0) = (a + v_n t)(b + v_n t)/(ab)$, where $a$ and $b$ are the initial major and minor axes of the tissue. This model fits our data well assuming the same edge speed $v_n \simeq 29.5$ µm/h for all tissues (*Figure 1E*). This observation suggests that edge speed is mostly independent of edge curvature. However, we measure a smaller edge speed at the major axes of ellipses, which are high-curvature points with radius of curvature $r_c \lesssim 0.75$ mm (*Figure 1—figure supplement 2*). Such high curvatures are concentrated around the major axes of our elliptical tissues. However, most of the tissue edge has a smaller curvature, and therefore advances at a curvature-independent speed. Further, even high curvature regions blunt due to expansion over time (see *Figure 1—video 3*). As a result, our model with a single edge speed $v_n \simeq 29.5$ µm/h is sufficient to capture the area expansion of both circular and elliptical tissues (*Figure 1E*).

Together, our findings demonstrate that epithelial shape and size determine area expansion dynamics via the perimeter-to-area ratio. This relationship results from the fact that tissues exhibit a constant, size-independent, migration-driven edge speed normal to tissue boundary. Since initial tissue size does not affect boundary dynamics, but does impact the relative growth and expansion of the tissue, we hypothesize that cells in the tissue bulk exhibit tissue size-dependent behaviors.

## Spatiotemporal dynamics of migration speed and radial velocity

Having demonstrated the role of the boundary in the expansion of large-scale epithelia, we sought to relate tissue areal expansion rate to internal collective cell migration dynamics. We used Particle-Image-Velocimetry (PIV, Materials and methods) to obtain flow fields describing cell migration within freely expanding epithelia (*Poujade et al., 2007*; *Petitjean et al., 2010*; *Angelini et al., 2010*; *Cohen et al., 2014*; *Aoki et al., 2017*). We constructed kymographs (Materials and Methods) to display the full spatiotemporal flow patterns of the tissue (*Figure 2A,B*; *Serra-Picamal et al., 2012*; *Zhang et al., 2017*), averaging over the angular direction and over 16 tissues (for representative kymographs, see *Figure 2—figure supplement 1*). We also separately show time evolution (*Figure 2C*) and spatial profiles (*Figure 2D*) of speed and radial velocity to compare small and large tissues.

Kymographs of speed and radial velocity reveal the existence of an edge region of fast, outward, radial cell motion (*Figure 2A,B*), with speeds similar to the radial edge velocity reported in *Figure 1D*. Up to ~500 µm from the tissue edge, the speed and radial velocity profiles are practically identical for small and large tissues (*Figure 2D*), showing that cell motion near the tissue edge is independent of tissue size.

The tissue centers, in contrast, exhibit size-dependent behaviors. For both small and large tissues, a wave front of cell speed and radial velocity propagates toward the tissue centers at ~90 µm/h (*Figure 2A and B*, dashed lines). This is approximately 3X faster than the tissue edge speed, consistent with previously described waves of strain rate in cell monolayers (*Serra-Picamal et al., 2012*). Soon after the wave of radial velocity reaches the center, it retreats, leaving a region of low radial velocity that increases in extent in the center of both small and large tissues (*Figure 2B*). This

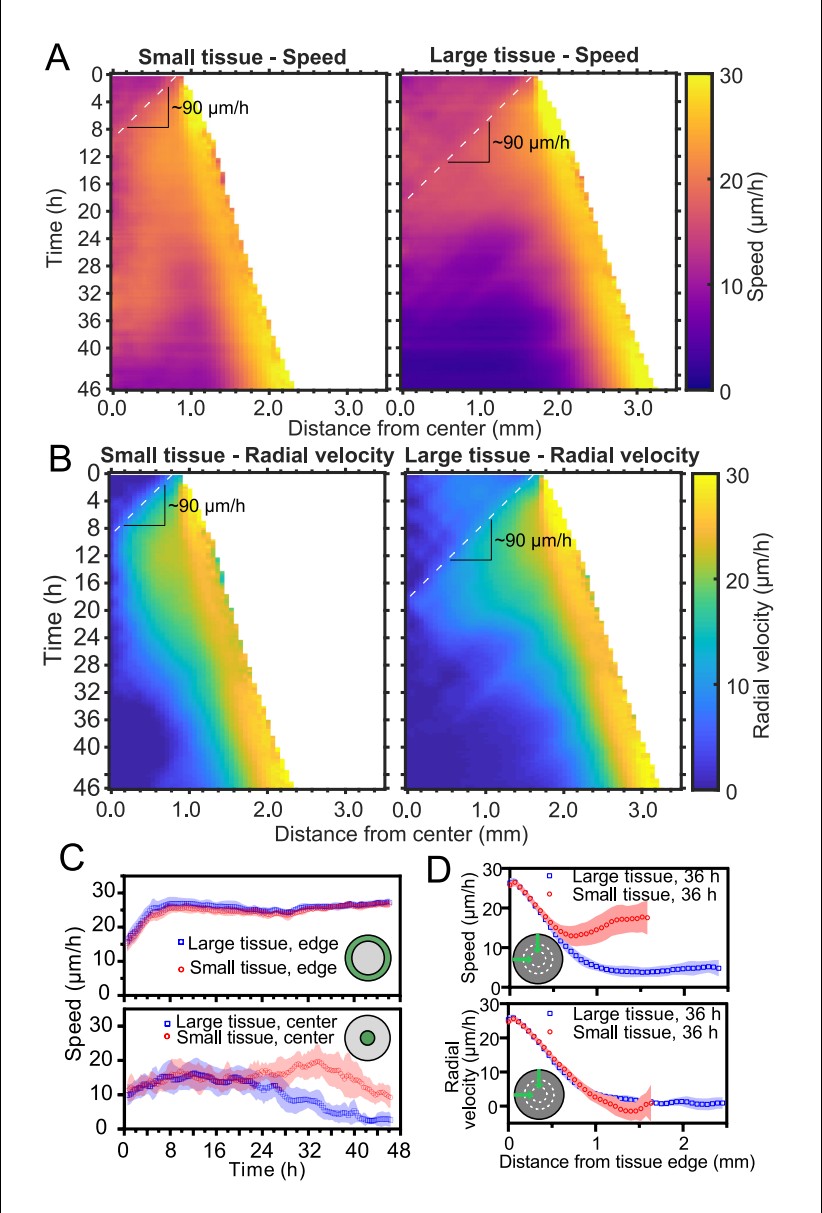

**Figure 2.** Speed and radial velocity in inner and outer tissue zones. (**A,B**) Average kymographs of (**A**) speed and (**B**) radial velocity $v_r$ throughout expansion for small (left) and large (right) tissues. (**C**) Evolution of the average speed of boundary (top) and center (bottom) zones, defined as regions extending ~200 μm from the tissue center and tissue edge, respectively. This width of the zones corresponds approximately to the velocity-velocity correlation length for MDCK cells (**Petitjean et al., 2010**). While the speed in the edge zone remains high in both small and large tissues, the speed in the center zone begins to decrease ~24 hr sooner in large tissues than in small tissues, as the central zone of the small tissues has particularly high speed from 18 to 36 hr. (**D**) Profiles of speed (top) and radial velocity (bottom) at 36 hr, from the edge of the tissue inwards. Arrows indicate that the tissues are indexed from the edge of the tissue inwards. All data are from n = 16 tissues across five independent experiments (small and large circles). Speed and radial velocity profiles of large and small tissues match closely for the first 500 μm from the tissue edge. The average difference between the profiles in this zone is 0.39 μm/h (speed) and 0.27 μm/h (radial velocity), respectively, while the smallest standard deviation for any point in either profile is 0.56 μm/h.

The online version of this article includes the following figure supplement(s) for figure 2:

**Figure supplement 1.** Representative kymographs and heatmaps for speed and radial velocity.

decrease of radial velocity is accompanied by a reduction in cell speed in the center of large tissues but not in small tissues, in which cell speed remains high until 36 hr (*Figure 2A and C* Bottom). We examine the behavior of this high-speed but low-radial-velocity central region of small tissues in the next section.

## Emergence of large-scale vortices

The propagation of low radial velocity out from the center of small tissues coincides with the formation and expansion of a millimeter-scale, persistent vortex (see *Figure 3A*, *Figure 3—video 1* for representative vortex). These large vortices are observed in both small and large tissues (*Figure 3—video 2*), but they only reach tissue-spanning sizes in small tissues.

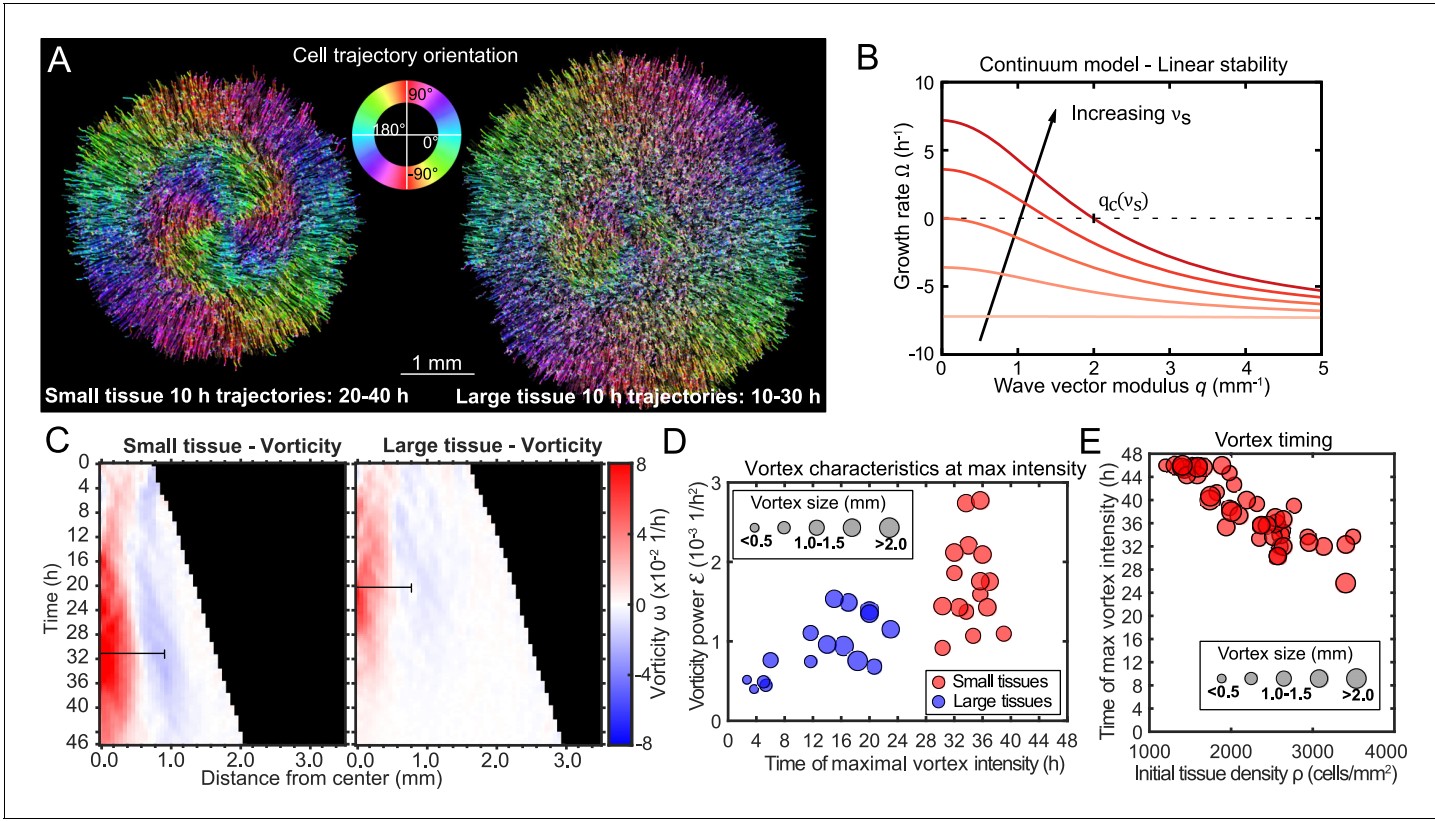

**Figure 3.** Vortex formation in expanding tissues. (A) Vortical flows seen from 10 hr traces of cell trajectories in small (left) and large (right) tissues. We color each trajectory according to its local orientation. (B) Growth rate of perturbations of wave vector modulus $q$ around the unpolarized state of the tissue bulk, *Equation 3*. Perturbations with wavelength longer than $2\pi/q_c$ grow ($\Omega > 0$), leading to large-scale spontaneous flows in the tissue bulk. We show curves for the following values of the polarity-velocity coupling parameter: $\nu_s = 0, 1, 2, 3, 4$ mm$^{-1}$. For the remaining parameters, we took $T_a = 100$ Pa/μm, $\xi = 100$ Pa·s/μm$^2$, $\eta = 25$ MPa·s, $\gamma = 10$ kPa·s, $a = 20$ Pa, $K = 10$ nN, as estimated in *Pérez-González et al., 2019*. (C) Average kymographs of vorticity show that the vortex in small tissues appears in the center and expands to >1 mm (n = 16), while vorticity in large tissues is present only during the early stages of tissue expansion (n = 16). The black bars indicate a characteristic vortex size. (D) Characteristic vortex size (marker size), time (horizontal axis), and intensity (vertical axis) of each tissue's maximal vortex intensity. Small tissue vortices are generally more intense, with $p < 0.0001$. (E) For small tissues, the time of maximal vortex intensity decreases with the initial cell density.

The online version of this article includes the following video and figure supplement(s) for figure 3:

**Figure supplement 1.** Trajectory displacement quantification of vortex.
**Figure supplement 2.** Representative kymographs and heatmaps of vorticity.
**Figure supplement 3.** Kymographs of enstrophy for small tissues of varying density and large tissues.
**Figure 3—video 1.** Vortex formation in a sample small expanding tissue from t = 22 hr to 40 hr of expansion.
https://elifesciences.org/articles/58945#fig3video1
**Figure 3—video 2.** Vortex formation in a sample large expanding tissue from t = 12 hr to 30 hr of expansion.
https://elifesciences.org/articles/58945#fig3video2

To visualize the form and scale of these vortices, we tracked individual cell motion and colored cell trajectories according to their orientation (*Püspöki et al., 2016*) for a representative small and large tissue vortex (see *Figure 3A* and Materials and Methods). We plotted trajectories for the time periods that the vortex was most apparent, which was 20–40 hr in the small tissue (*Figure 3A*, left) and 10–30 hr in the large tissue (*Figure 3A*, right). During the vortex period in small tissues, cell trajectories are primarily radial in the boundary zone, but mainly tangential in the entire central zone (*Figure 3A* left, see *Figure 3—figure supplement 1* for vortex trajectory quantification).

To understand the emergence of the vortices, we build on a continuum physical model of tissue spreading that describes the cell monolayer as a two-dimensional compressible active polar fluid (*Blanch-Mercader et al., 2017*; *Pérez-González et al., 2019*; *Alert et al., 2019*). Consistent with our velocity measurements (*Figure 2C*), we assume that cells at the edge zone are radially polarized and motile, whereas cells in the bulk of the tissue are unpolarized and non-motile. We describe cell polarization at a coarse-grained level via a polarity field $\mathbf{p}$ that obeys the following dynamics (*Alert and Trepat, 2020*):

$$\partial_t \mathbf{p} = \frac{\mathbf{h}}{\gamma} + \nu_s \mathbf{v}. \tag{1}$$

Here, $\gamma$ is the rotational viscosity that damps polarity changes. Respectively, $\mathbf{h} = -a\mathbf{p} + K\nabla^2\mathbf{p}$ is the so-called molecular field that governs polarity relaxation: the first term drives the polarity to zero, and the second term opposes spatial variation of the polarity field. As a result of these terms, the radial polarity at the tissue edge decays over a length scale $L_c = \sqrt{K/a}$ into the tissue bulk.

With respect to previous models of tissue spreading, we add the last term in *Equation 1*, which couples the polarity to the tissue velocity field $\mathbf{v}$. This coupling is a generic property of active polar fluids interacting with a substrate (*Brotto et al., 2013*; *Kumar et al., 2014*; *Oriola et al., 2017*; *Maitra et al., 2020*). Previous works in agent-based models showed that similar polarity-velocity alignment interactions (*Alert and Trepat, 2020*) can lead to waves (*Petrolli et al., 2019*), flocking transitions (*Szabó et al., 2006*; *Henkes et al., 2011*; *Basan et al., 2013*; *Malinverno et al., 2017*; *Giavazzi et al., 2018*), and vortical flows (*Rappel et al., 1999*; *Camley et al., 2014*; *Li and Sun, 2014*; *Segerer et al., 2015*; *Barton et al., 2017*; *Lin et al., 2018*) in small, confined, and polarized tissues. Here, using a continuum model, we propose that cell polarity not only aligns with but is also generated by tissue flow, and we ask whether this polarity-velocity coupling can lead to large-scale spontaneous flows in the unpolarized bulk of unconfined tissues.

To determine the flow field $\mathbf{v}$, we impose a balance between internal viscous stresses in the tissue, with viscosity $\eta$, and external cell-substrate forces, including viscous friction with coefficient $\xi$, active traction forces with coefficient $T_a$, and the cell-substrate forces associated with the polarity-velocity coupling $\nu_s$:

$$\eta\nabla^2\mathbf{v} = \xi\mathbf{v} - T_a\mathbf{p} - \nu_s\mathbf{h}. \tag{2}$$

This force balance predicts that even if cell polarity, and hence active traction forces, are localized to a narrow boundary layer of width $L_c \sim 50$ μm (*Blanch-Mercader et al., 2017*; *Pérez-González et al., 2019*), cell flow can penetrate a length $\sim\lambda = \sqrt{\eta/\xi}$ into the tissue. Based on our measurements (*Figure 2D*), we estimate $\lambda \sim 0.5 - 1$ mm, which is larger than the velocity correlation length of $\sim 200$ μm in the tissue bulk (*Petitjean et al., 2010*).

A linear stability analysis of *Equations 1 and 2* shows that perturbations of wave number $q$ around the quiescent ($\mathbf{v} = 0$) and unpolarized ($\mathbf{p} = 0$) state grow with a rate

$$\Omega(q) = -\frac{a}{\gamma}(1 + L_c^2 q^2) + \frac{T_a\nu_s - a\nu_s^2(1 + L_c^2 q^2)}{\xi(1 + \lambda^2 q^2)}. \tag{3}$$

This result shows that, if $T_a\nu_s > a(\xi/\gamma + \nu_s^2)$, the unpolarized state of an active polar fluid described by *Equations 1 and 2* is unstable ($\Omega > 0$) to perturbations of wavelength longer than a critical value $2\pi/q_c$ given by $\Omega(q_c) = 0$ (*Figure 3B*). This analysis suggests that, for tissues larger than this critical value $\sim 2\pi/q_c$, the quiescent tissue bulk becomes unstable and starts to flow spontaneously at large scales, consistent with the emergence of large-scale vortices. The mechanism of this instability is the positive feedback between flow-induced cell polarization and the flows due to migration of polarized

cells. The fact that a critical size of the order of millimeters is required for this long-wavelength instability might explain why large-scale vortices have not been observed in previous studies, which considered smaller tissues.

## Vortex kinematics

To quantify the kinematics of the large-scale vortical flows, we obtained the vorticity field $\omega(\mathbf{r}, \mathbf{t}) = \nabla \times \mathbf{v}(\mathbf{r}, \mathbf{t})$. Before averaging over tissues, we took the dominant direction of rotation of each tissue to correspond to positive vorticity. This direction was counterclockwise in 51.5% of tissues and clockwise in 49.5% of tissues, with a sample size of 68. With this convention, the vortex core always has positive vorticity. Accordingly, the outer region of the vortex exhibits negative vorticity (*Figure 3C*, see *Figure 3—figure supplement 2* for kymographs and heatmaps of vorticity representative tissues), which corresponds to the counter-rotation that occurs when the central vortical flow transitions to the outer radial flow (*Figure 3A*, left). We define a characteristic vortex radius as the radial position of the center of the negative-vorticity region, which is ~1 mm at 36 hr in small tissues (*Figure 3C*, black bars).

To analyze vortex dynamics across different tissues with varying vortex positioning, and to quantitatively capture the onset and strength of vortices, we calculated the enstrophy spectrum $\mathcal{E}(q, t) = |\tilde{\omega}(\mathbf{q}, t)|^2$, where $\tilde{\omega}(\mathbf{q}, t) = \int (d\mathbf{r}/A)\omega(\mathbf{r}, t)e^{i\mathbf{q}\cdot\mathbf{r}}$ are the spatial Fourier components of the vorticity field $\omega(\mathbf{r}, t)$ (*Alert et al., 2020*). The enstrophy spectrum is the power spectral density of the vorticity field as a function of the wave-vector modulus $q$, and therefore provides a measure of the vortex intensity at a length scale $2\pi/q$. The kymographs of the enstrophy spectrum show that most of the vortex's intensity is found at a characteristic length scale of ~1 mm (*Figure 3—figure supplement 3*).

For each tissue we characterized the maximal vortex strength by the maximum value of $\mathcal{E}(q, t)$ as well as its associated wavelength $2\pi/q$ and time of occurrence. We represented these three quantities on a scatter plot, which shows that vortices in small tissues have generally higher intensity than those in large tissues (*Figure 3D*). Vortices in small tissues are also larger relative to tissue size, since the absolute size of vortices in small and large tissues is similar (*Figure 3D*). Furthermore, vortex strength peaks several hours later in small tissues than in large tissues (*Figure 3D*). We hypothesized that this difference is due to large tissues featuring a faster density increase than small tissues (*Figure 1C*). To test this hypothesis, we varied the initial cell density of small tissues and observed that the time of maximum vortex intensity decreases with increasing density (*Figure 3E*, *Figure 3—figure supplement 3*). These results prompted us to examine spatiotemporal cell density evolution.

## Spatiotemporal dynamics of cell density

Given that cell density appears to affect vortex formation and is known to control contact inhibition of locomotion and proliferation (*Schnyder et al., 2020*), we explored the spatiotemporal evolution of cell density. Constructing average kymographs in the same way as for speed, radial velocity, and vorticity, we observe that the vortex region in the center of small tissues is accompanied by an unexpected local density minimum (*Figure 4A*). Strikingly, snapshots of small and large tissues reveal that large-scale vortices occur in low-density regions, regardless of location within the tissue (*Figure 4—figure supplement 1*). However, given that vortices in large tissues are often off-centered, the low-density region does not appear in their average kymograph of cell density (*Figure 4A*).

To investigate the effects of initial conditions, we tracked the density evolution of the center and boundary zones across tissues with different starting densities and sizes, grouping initial densities into three ranges as before (*Figure 4B and C*). As with the average density in *Figure 1C*, the density monotonically increases in large tissues centers but is non-monotonic in small tissues. Notably, the cell density at the center of small tissues of different initial cell densities reach a common minimum during the 16–32 hr time period (*Figure 4B*), which includes the vortex onset time. At the boundary zone, the long-time evolution of the cell density is independent of initial tissue size and density (*Figure 4C*). This common long-time evolution is reached at about 12 hr (*Figure 4C*), which coincides with the time at which the edge radial velocity stabilizes upon the overshoot (*Figure 1D*).

To understand the unexpected transient density decrease at the center of small tissues, we sought to explain it as the result of combined advective transport based on the measured radial flow

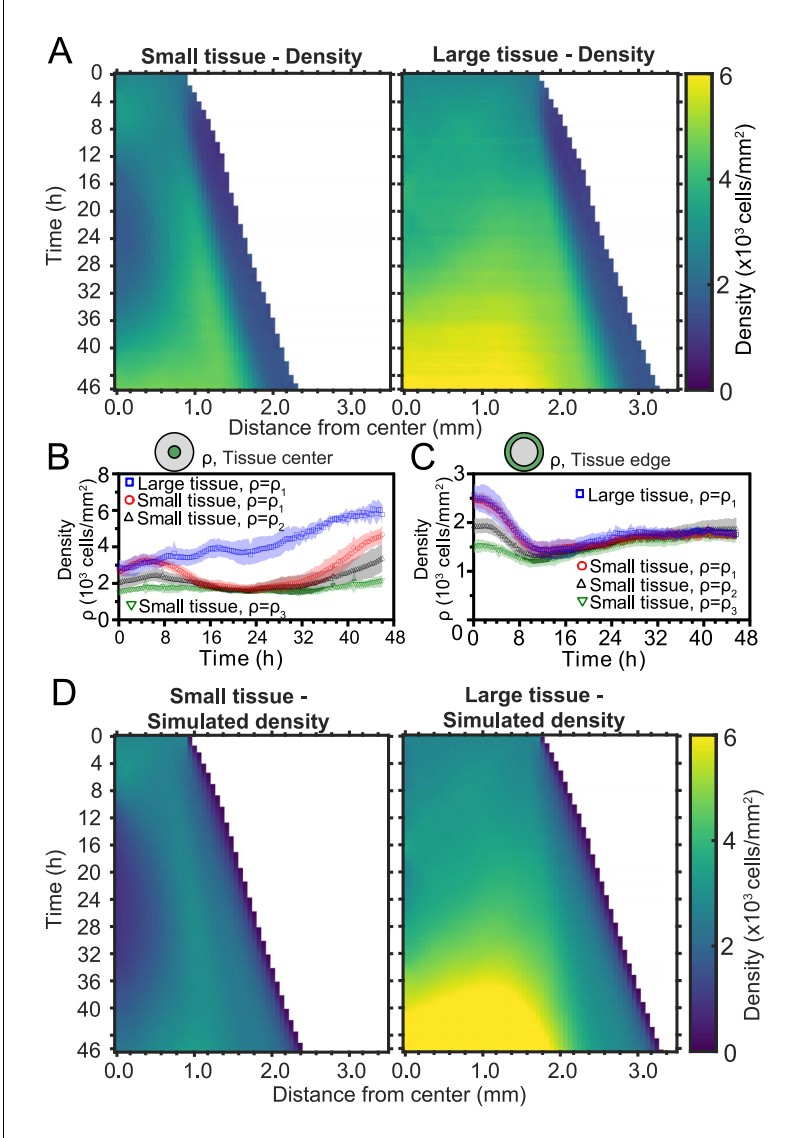

**Figure 4.** Spatiotemporal dynamics of cell density during epithelial expansion. (A) Averaged kymographs of cell density for small (left, n = 11) and large (right, n = 9) tissues. Small tissues develop a central low-density region that persist more than 20 hr. (B) Cell density, ρ, at the center of large tissues increases gradually, while cell density at the center of small tissues has non-monotonic evolution. (C) For different initial tissue sizes and densities, the evolution of the cell density, ρ, at the boundary zone converges to similar values at about 12 hr, which coincides with the end of the overshoot of edge radial velocity in *Figure 1D*. Center and boundary zones are defined as in *Figure 2B*. (D) Simulated evolution of cell densities obtained from the numerical solution of the continuity equation using the average radial velocity measurements $v_r(r,t)$ (*Figure 2B*) and a uniform and constant cell proliferation rate corresponding to a 16 h cell doubling time. In (B,C) the initial cell density ranges and number of replicates $\rho_1$, $\rho_2$, and $\rho_3$ are the same as in *Figure 1D*.

The online version of this article includes the following figure supplement(s) for figure 4:

**Figure supplement 1.** Large vortices co-occur with regions of low cell density.

---

fields $\mathbf{v}_r(\mathbf{r},t)$ and homogeneous cell proliferation at a rate $k(\mathbf{r},t) = k_0$ throughout the tissue. To test this hypothesis, we solved the continuity equation for the cell density field $\rho(\mathbf{r},t)$,

$$\frac{\partial \rho}{\partial t} = -\nabla \cdot (\rho \mathbf{v}) + k_0 \rho, \tag{4}$$

using the average radial velocity profiles $v_r(r,t)$ measured by PIV (*Figure 2D*), and a proliferation rate $k_0 = 1.04$ h$^{-1}$, which corresponds to a cell doubling time of 16 hr (Materials and methods). This minimal model recapitulates the major features of the evolving density profiles for both small and large tissues (compare *Figure 4D* with *Figure 4A*). Therefore, the unexpected formation of a central low-density region results from the combination of outward tissue flow and proliferation within the colony. However, further research is required to determine the biophysical origin of the non-monotonic density evolution. Moreover, having assumed a density-independent proliferation rate, our model predicts a cell density in the center of large tissues higher than the one measured at the end of the experiment, and it does not quantitatively reproduce the cell density profiles at the edge regions. These discrepancies suggest that more complex cell proliferation behavior is required to fully recapitulate the density dynamics in expanding cell monolayers.

## Spatiotemporal dynamics of cell cycle

To better understand how tissue expansion affects cell proliferation, we analyzed the spatiotemporal dynamics of cell-cycle state. Our cells stably express the FUCCI markers, meaning that cells in the G0-G1-S phase of the cell cycle (referred to here as G1) fluoresce in red (shown as magenta), and cells in the S-G2-M phase of the cell cycle (referred to here as G2) fluoresce in green (*Sakaue-Sawano et al., 2008*). Additionally, immediately-post-mitotic cells do not fluoresce and appear dark. Small and large tissues are initially well mixed with green and magenta cells, confirming that cells are actively cycling throughout the tissue at the time of stencil removal (*Figure 5—figure supplement 1*). During tissue expansion, spatiotemporal patterns of cell-cycling behavior emerge (*Figure 5A*, *Figure 5—video 1*).

To quantitatively investigate these cell-cycle patterns, we obtained the local fractions of G1, G2, and post-mitotic cells by evaluating cell cycle state for each cell nucleus (see Materials and Methods). We then overlaid kymographs of the G1 and G2 cell-cycle-state fractions (*Figure 5B*) and plotted the time evolution of G1, G2, and post-mitotic fractions together (*Figure 5C,D*). Immediately after stencil removal, we observe a cell division pulse in all tissues, which manifests in a decrease in G2 and increase in post-mitotic fraction (*Figure 5C,D*). After about 12 hr of tissue expansion, the boundary region becomes primarily populated by rapidly-cycling cells (*Figure 5B,C*), which results in a predominance of cells in this region that either have recently divided (post-mitotic, black) or are likely to divide soon (G2, green). The high numbers of post-mitotic cells indicate that cells in G1 rapidly proceed to mitosis. Given that the edge radial speed overshoots during the first 12 hr of tissue expansion (*Figure 1D*), future work is necessary to characterize the effect of cell cycling on edge motion at early stages of expansion.

In the central region of small tissues (*Figure 5B* left, D left), we observe cell-cycling dynamics similar to the boundary region. Thus, in the tissue-spanning vortex of small tissues, cells are also rapidly cycling. The fraction of cells in G1 only starts to increase at ~40 hr (*Figure 5D* left), coinciding with the weakening of the vortex (*Figure 3C* left). In contrast, the center zone of large tissues undergoes strong cell-cycle arrest at the G1-G2 transition at about 30 hr, also coinciding with the weakening of the vortex in large tissues (*Figure 5B* right, D right). Cells already past G1 at this time continue to division and re-enter G1, evidenced by the steady increase in local fraction of G1 accompanied by a steady decrease in G2 after 30 hr. Similar cell-cycle arrests were previously reported both in growing epithelia (*Streichan et al., 2014*) and in spreading 3D cell aggregates (*Beaune et al., 2014*). Before the onset of cell-cycle arrest, the center of large tissues exhibits large-scale coordinated cell-cycling dynamics in the form of anti-phase oscillations, with peaks in G2 fraction accompanied by troughs in G1 fraction (*Figure 5B* right, D right).

Finally, we sought to link cell-cycle dynamics to the kinematics of tissue expansion by studying correlations between local measurements of cell cycle, cell speed, and cell density (*Figure 5E*). Here, each point represents one PIV window, with color indicating its average cell-cycle state. As expected, cell speed is negatively correlated with cell density. Further, in large tissues, the cell-cycle state transitions from G1-dominated to G2-dominated when cell density increases above ~5000 cells/mm$^2$ and cell speed falls below ~12 μm/h (*Figure 5E* right). In this regime, the decrease of cell speed with increasing cell density bears similarities to previously-reported glass transitions and contact inhibition of locomotion (*Angelini et al., 2011*; *Zimmermann et al., 2016*; *Garcia et al., 2015*). Small tissues, by contrast, lack the G1-dominated, slow, high-density cell population (*Figure 5E*, left) found in the center of large tissues. Taken together, our findings emphasize that cell cycling, cell

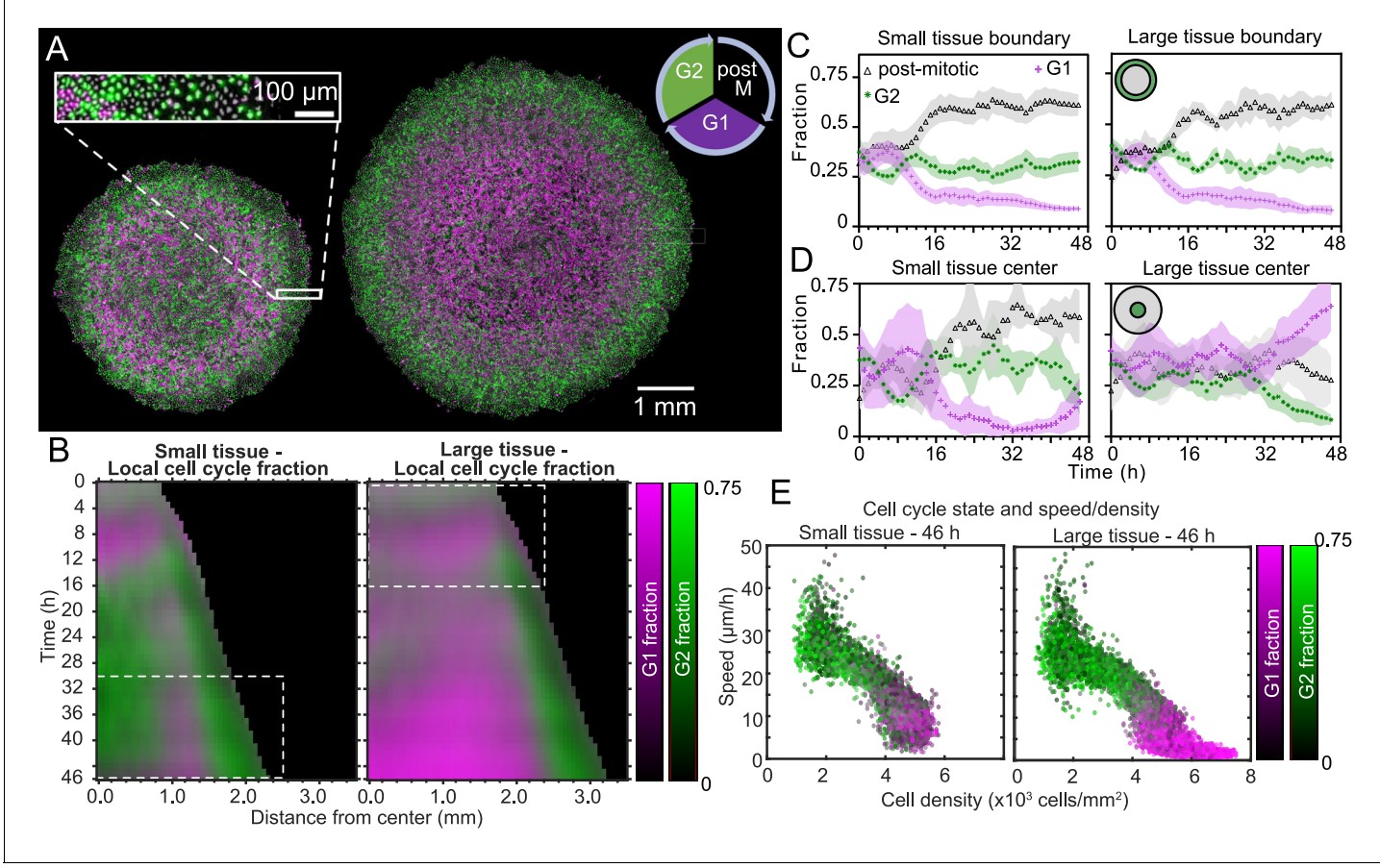

**Figure 5.** Coordinated spatiotemporal cell-cycle dynamics. Transition from the G1 (magenta) to the G2 (green) phase of the cell cycle corresponds to DNA replication (during S phase). Subsequently, a cell proceeds to mitosis (M phase, dark), and eventually back to the G1 phase upon cell division. (**A**) Fluorescence images of the Fucci marker of cell-cycle state at the end of the experiment (46 hr) of representative small and large tissues overlaid with nuclei positions (gray). The boundary zone of both tissues has more cells in the G2 than in the G1 phase, along with a substantial proportion of dark cells (inset). Scale bars 1 mm. (**B**) Average kymographs (small, n = 5; large, n = 11) of cell-cycle-state fraction. In small tissues, a G1-dominated transition zone, which appears as a vertical magenta streak from 16 hr onward, is interposed between G2-dominated center and edge zones. While the size of small tissues from 30 to 46 hr matches that of large tissues from 0 to 16 hr (dashed boxes), cell-cycle states between these times are clearly distinct. (**C**) Fraction of cell-cycle states in the boundary zone. (**D**) Fraction of cell-cycle states in the center zone. Center and boundary zones are defined as in *Figure 2*. For C and D, n = 5 for small tissues and n = 11 for large tissues. (**E**) Scatter plot of density and speed, with color indicating the fraction of cells at G1 and G2, corresponding to each PIV pixel of the final timepoint of a representative small (left) and large (right) tissue.

The online version of this article includes the following video and figure supplement(s) for figure 5:

**Figure supplement 1.** Snapshots of cell cycle at t = 0 hr show tissue cell cycle is well-mixed at the time of stencil removal.

**Figure 5—video 1.** Coordinated spatiotemporal cell-cycle dynamics in expanding monolayers.

https://elifesciences.org/articles/58945#fig5video1

flow, and cell density patterns are inextricably linked and depend on the initial size of an expanding tissue.

## Discussion

We began this study by asking how changes in initial size affect the long-term expansion and growth of millimeter-scale epithelia. By means of high spatiotemporal resolution imaging and precisely controlled initial conditions, our assays systematically dissected tissue expansion and growth from the overall boundary kinematics (*Figure 1*) to the internal flow patterns (*Figures 2*, *3* and *4*) and cell-cycle dynamics (*Figure 5*). While we demonstrated that 'small' tissues increase in area relatively much faster than do 'large' tissues, our data suggest a surprising and stark decoupling of the outer

and inner regions of an expanding epithelium. Notably, the behaviors of the edge zones are largely independent of tissue size, cell density, and history, while interior dynamics depend strongly on these factors.

Unexpectedly, the overall tissue growth and expansion dynamics (*Figure 1*) could be attributed to one dominant feature: these epithelia expanded at the same edge speed regardless of initial tissue size, shape, and cell density. The only exception is the major axes of ellipses, where the normal edge speed is smaller when the radius of curvature is $r_c<0.75$ mm. This observation, combined with the fact that the velocity penetration length is 500 mm (*Figure 2D*), suggests that a tissue must be 1 mm in diameter for the tissue edge to move independently of bulk flows. As a result of this robust edge motion, the areal expansion rate of the tissue is dictated by its perimeter-to-area ratio. To further emphasize the decoupling of the boundary and internal dynamics of epithelia, consider that the key findings in *Figure 1* neither predict nor depend upon the radically different internal dynamics we observed within 'small' and 'large' tissues. For instance, despite the roiling vortices occupying large portions of 'small' tissues and the pronounced, large-scale contact inhibition of 'large' tissues–two antithetical phenomena–no hints of these behaviors can be detected in the motion of the boundary.

Critically, the type and timing of internal dynamics are dictated not by the current size but by the expansion history of a given tissue. While a small tissue eventually expands to reach the initial size of a large tissue, it exhibits different internal dynamics from the large tissue at this size (*Figure 6*). This difference in internal dynamics is perhaps easiest to observe in spatiotemporal evolution of cell cycle (*Figure 6D*, *Figure 5B* dashed boxes); the small-tissue footprint from 30 to 46 hr closely matches the large-tissue footprint from 0 to 16 hr, but the cell cycle distribution during these time periods bears almost no similarities. This applies as well to other important bulk properties of the tissue (*Figure 6A–C*), as cell cycle is tightly linked to cell speeds and density (*Figure 5E*). For example, at equal current sizes, the center of initially-small tissues features high vorticity with decreasing cell speed whereas initially-large tissues exhibit low vorticity and increasing cell speed (*Figure 6A,B*). Respectively, at equal current sizes, while absolute cell densities in the tissue centers share some overlap, it is notable that the rate of density change at the tissue center is increasing faster in initially-small tissues than in initially-large tissues (*Figure 6C*). However, the most striking differences in cell density evolution occur not at equal current sizes but during the early stages of tissue expansion: whereas the cell density at the center of large tissues increases at all times, the center of small tissues features a marked density decrease between ~8 and ~24 hr (*Figure 4A,B*). Overall, while edge dynamics are stereotyped and conserved across different sizes, our findings suggest that initial tissue

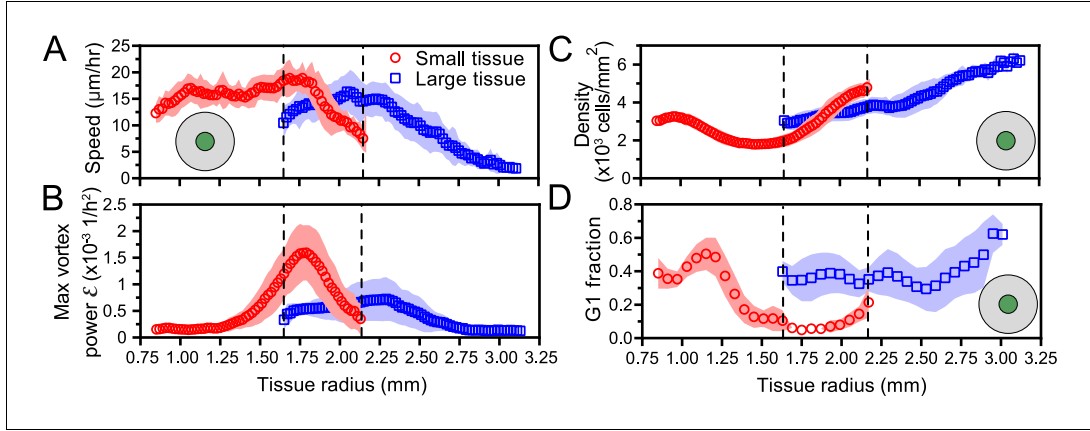

**Figure 6.** Initial tissue size, rather than current tissue size, determines the internal dynamics of expanding epithelia. Here, we quantify the internal state of the tissue in terms of the cell speed in tissue center (**A**), maximal vortex power (**B**), cell density in tissue center (**C**), and fraction of cells in the G1 phase of the cell cycle in tissue center (**D**). At late times, initially-small tissues reach radii that initially-large tissues had at early times. When they have the same current size (overlap region in between dashed lines), initially-small and initially-large tissues have distinct internal dynamics of cell migration and cell proliferation. The tissue center zone in **A**, **C**, and **D** was defined as in *Figure 2*.

size impacts the bulk dynamics by altering the constraints under which the tissue grows. We expect that tissues with sizes between our two choices would exhibit similar edge dynamics and internal patterns that cross over between our small and large tissues.

The vortices are a particularly striking example of such size- and history-dependent internal patterns (*Figure 3*, *Figure 6B*). Our active fluid model suggests that the vortices emerge from a dynamical instability of the tissue bulk, which occurs when the tissue reaches a critical size. Thus, whereas the instability itself is a bulk phenomenon independent of the tissue edge, edge-driven expansion allows small tissues to reach the critical size that triggers the instability. In addition, our data suggest a strong correlation between vortex formation and the development of non-monotonic density profiles. Not only did small tissues exhibit co-occurrence of vortices with density decreases in the tissue center, but also off-center vortices in large tissues always co-localized with a local density decrease (*Figure 4—figure supplement 1*). Our model does not currently describe cell density, and hence cannot explain the relationship between vortex formation and local density decreases. Thus, our experimental findings call for the development of more detailed models that couple cell density to both the velocity and the polarity fields, accounting for how density gradients influence cell polarization (*Alert and Trepat, 2020*).

The pronounced decoupling between boundary and internal dynamics in epithelia confers stability to the overall expansion of the tissue, making it robust to a wide range of internal perturbations. From the perspective of collective behavior, we speculate that such robust boundary dynamics may be beneficial in a tissue such as an epithelium whose teleology is to continuously expand from its free edges to sheath organ surfaces. Further, the ability to accurately predict epithelial expansion with a single parameter, the edge speed, will have practical uses in experimental design and tissue-engineering applications. Finally, given that many of the phenomena presented here only occurred due to the millimetric scale of our unconfined tissues and the long duration of the experiments, our results showcase the value of pushing the boundaries of large-scale, long-term studies on freely-expanding tissues.

# Materials and methods

**Key resources table**

| Reagent type (species) or resource | Designation | Source or reference | Identifiers | Additional information |
|---|---|---|---|---|
| Cell line (canine) | MDCK-FUCCI | *Streichan et al., 2014*. Wildtype: ECACC-00062107 | N/A | RFP signal is G1, GFP is G2. |
| Other | DMEM - low glucose | Sigma-Aldrich, Inc | Cat.D5523 | |
| Other | fetal bovine serum | Atlanta Biologicals | Cat.S11550 | |
| Other | Silicone stencil material, 250 µm thick. | Stockwell Elastomerics | Cat.HT6240-40D | Tissue patterning material |
| Software, algorithm | FIJI | NIH ImageJ Project | | |
| Software, algorithm | MATLAB | Mathworks, Inc | 2019A | All code compatible back to at least 2015B |
| Software, algorithm | Machine Learning Tools | Laboratory code *LaChance and Cohen, 2020* | —see References | Full code on GitHub |

## Cell culture

All experiments were performed with MDCK-II cells expressing the FUCCI cell-cycle marker system as received from: *Streichan et al., 2014*. After treatment with Mycoplasma Removal Agent (MPI Biological), cells tested negative for mycoplasma (MycoProbe, R and D Systems). We cultured cells in MDCK media consisting of low-glucose (1 g/L) DMEM with phenol red (Gibco, USA), 1 g/L sodium bicarbonate, 1% streptomycin/penicillin, and 10% FBS (Atlanta Biological, USA). Cells were maintained at 37°C and 5% $CO_2$ in humidified air.

## Tissue patterning

We coated tissue-culture plastic dishes (BD Falcon, USA) with type-IV collagen (MilliporeSigma, USA) by incubating 150 μL of 50 μg/mL collagen on the dish under a glass coverslip for 30 min at 37°C, washing three times with deionized distilled water (DI), and allowing the dish to air-dry. We then fabricated silicone stencils with cutouts of desired shape and size and transferred the stencils to the collagen coated surface of the dishes. Stencils were cut from 250 μm thick silicone (Bisco HT-6240, Stockwell Elastomers) using a Silhouette Cameo vinyl cutter (Silhouette, USA). We then seeded the individual stencils with cells suspended in media at 1000 cells/mL. Suspended cells were concentrated at $\sim 2.25 \times 10^6$ cells/mL and pipetted cells into the stencils at the appropriate volume. Care was taken not to disturb the collagen coating with the pipette tip. To allow attachment of cells to the collagen matrix, we incubated the cells in the stencils for 30 min in a humidified chamber before flooding the dish with media. We then incubated the cells for an additional 18 hr to allow the cells to form monolayers in the stencils, after which the stencils were removed with tweezers. Imaging began 30 min after stencil removal. Media without phenol red was used throughout seeding and imaging to reduce background signal during fluorescence imaging.

## Live-cell time-lapse imaging

All imaging was performed with a 4X phase contrast objective on an automated, inverted Nikon Ti2 with environmental control (37°C and humidified 5% CO2) using NIS Elements software and a Nikon Qi2 CMOS camera. Phase contrast images were captured every 20 min, while RFP/GFP channels were captured every 60 min at 25% lamp power (Sola SE, Lumencor, USA) and 500 ms exposure time. No phototoxicity was observed under these conditions for up to 48 hr. Final images were composited from 4 × 4 montages of each dish using NIS Elements.

## Tissue edge radial velocity

Tissues were segmented to make binary masks using a custom MATLAB (Mathworks) script. Tissue edge radial velocity was measured from the binary masks within more than 200 discrete sectors of the tissue; the edge radial velocity of all sectors were averaged to arrive at the tissue average edge radial velocity. Radial velocity at each sector was calculated for each timepoint as the rate of change of the average extent of the boundary pixels of the sector, utilized a rolling average of 3 timepoints (1 hr) to account for capture phase offsets resulting from capturing phase and fluorescence images at different frequencies. Sectors originated from the center of each tissue at the initial timepoint and were ~20 μm wide at the edge of the tissue at the starting point.

## Radius of curvature for the major and minor axes of elliptical tissues

Curvature at the major and minor axes of growing tissues was approximated at each time-point by fitting an ellipse to the tissue footprint and taking the radius of curvature at the minor and major axes as $b^2/a$ and $a^2/b$, respectively, where $a$ is the major semi-axis length and $b$ is the minor semi-axis length.

## Statistical tests and goodness of fit

Normalized $\chi^2$ values in *Figure 1E* were calculated as $\frac{1}{N}\sum_{i=1}^{N}\frac{(u_i-\mu_i)^2}{\sigma_i^2}$, where N is the number of timepoints in the curve, $u_i$ are the model predictions, and $\mu_i$ and $\sigma_i$ are the mean and standard deviation of the measured values, respectively. With these definitions, a fit with $\chi^2<1$ is good.

The P-value in *Figure 3D* was calculated using a Mann-Whitney U test, and the two-tailed p-value of $p<10^{-4}$ indicates that the large and small vortex power data indeed come from different populations.

## Cell counts

The FUCCI system contains a period after M-phase where cells go dark, making FUCCI unreliable for cell counting. Instead, we developed and trained a convolutional neural network to reproduce nuclei from 4X phase contrast images using our in-house Fluorescence Reconstruction Microscopy tool (*LaChance and Cohen, 2020*) . The output of this neural network was then segmented in ImageJ to determine nuclei footprints and centroids.

## Tissue PIV and density measurements

Tissue velocity vector fields were calculated from $2 \times 2$ resized phase contrast image sequences using the free MATLAB package PIVLab (*Thielicke and Stamhuis, 2014*) with the FFT window deformation algorithm. We used a 1 st pass window size of $64 \times 64$ pixels and second pass of $32 \times 32$ pixels, with 50% pixel overlaps. This resulted in a $115 \times 115$ μ*m* window. The window size was chosen to be smaller than the velocity-velocity correlation length but large enough to enable fast computation of PIV fields for many tissues. As seen in *Figure 2—figure supplement 1*, using a window size of $57 \times 57$ μ*m*, which contains only a few cells, yields higher resolution velocity fields but does not qualitatively affect the measured speed and radial velocity. We focus on large-scale features of the velocity field, which are not affected by choosing a smaller PIV window size.

Local density was also calculated for each PIV window by counting the number of approximate nucleus centroids in that window. Data from PIV were smoothed in time with a moving average of 3 time points centered at each timepoint as before.

## Average kymographs

First, we constructed kymographs for individual tissues using distance from the tissue center as the spatial index for each measurement window corresponding to a kymograph pixel. We did not plot kymograph pixels for which more than 95% of the measurements at that distance were beyond the tissue footprint. We then averaged the individual tissue kymographs, aligning by the centers.

## Trajectory colorization

We first generated a plot of all relevant trajectories (*Tinevez et al., 2017*) colorized randomly in grayscale using a custom MATLAB (Mathworks) script. We then used the Fiji plugin OrientationJ on this plot to colorize the resulting image according to orientation (*Püspöki et al., 2016*).

## Cell density simulation

To test whether the observed spatiotemporal evolution of density $\rho(r,t)$ could be explained by flow of material (rather than divisions, extrusions, and cell death), we solved the continuity equation for a homogenous tissue in a circular geometry with spatiotemporal evolution of average radial velocity $v_r(r,t)$ as measured from PIV in experiments (*Figure 2B*). The continuity equation is

$$\frac{\partial \rho}{\partial t} = -\nabla \cdot \mathbf{j} + k_0 \rho, \tag{5}$$

where a homogeneous cell proliferation rate $k_0 = 1.04 h^{-1}$ is assumed throughout the tissue, which corresponds to the cell doubling time of 16 hr. The current density is $\mathbf{j} = \rho \mathbf{v_r} - D\nabla\rho$, where we included a diffusion term with a small diffusion constant $D = 0.22\,\mathrm{mm^2/h}$ for numerical stability.

The continuity *Equation (5)* was discretized using the finite volume method (*Eymard et al., 2000*), which is briefly summarized below. The tissue domain was divided into an inner circle $\Omega_0$ of radius $r_{1/2} = \frac{1}{2}\Delta r$ and circular annuli $\Omega_i$ with inner radii $r_{i-1/2} = (i - \frac{1}{2})\Delta r$ and outer radii $r_{i+1/2} = (i + \frac{1}{2})\Delta r$, respectively, where $i = 1, 2, 3, \ldots$ and $\Delta r = 115 \mu m$ corresponds to the width of 1 window in the PIV analysis. The continuity *Equation (5)* was then integrated over the inner circle $\Omega_0$ and circular annuli $\Omega_i$ as

$$\frac{1}{A_0} \int_0^{r_{1/2}} (2\pi r dr) \frac{\partial \rho}{\partial t} = \frac{1}{A_0} \int_0^{r_{1/2}} (2\pi r dr) \left[ -\nabla \cdot \mathbf{j} + k_0 \rho \right], \tag{6a}$$

$$\frac{1}{A_i} \int_{r_{i-1/2}}^{r_{i+1/2}} (2\pi r dr) \frac{\partial \rho}{\partial t} = \frac{1}{A_i} \int_{r_{i-1/2}}^{r_{i+1/2}} (2\pi r dr) \left[ -\nabla \cdot \mathbf{j} + k_0 \rho \right], \tag{6b}$$

where $A_0 = \pi r_{1/2}^2$ is the area of the inner circle $\Omega_0$ and $A_i = \pi r_{i+1/2}^2 - \pi r_{i-1/2}^2$ is the area of the circular annulus $\Omega_i$. The integrals in *Equation (6a, b)* can be approximated as

$$\frac{\partial \rho(0,t)}{\partial t} = -\frac{2\pi}{A_0} r_{1/2} j(r_{1/2},t) + k_0 \rho(0,t), \tag{7a}$$

$$\frac{\partial \rho(r_i,t)}{\partial t} = -\frac{2\pi}{A_i} \left[ r_{i+1/2} j(r_{i+1/2},t) - r_{i-1/2} j(r_{i-1/2},t) \right] + k_0 \rho(r_i,t). \tag{7b}$$

Here, density profiles $\rho(r_i,t)$ are evaluated at $r_i = i\Delta r$ for all $i = 0,1,2,\ldots$. Current densities are evaluated as $j(r_{i+1/2},t) = \rho(r_{i+1/2},t)v_r(r_{i+1/2},t) - D[\rho(r_{i+1},t) - \rho(r_i,t)]/\Delta r$ for all $i = 0,1,2,\ldots$, where $\rho(r_{i+1/2},t) = [\rho(r_i,t) + \rho(r_{i+1},t)]/2$ and $v_r(r_{i+1/2},t) = [v_r(r_i,t) + v_r(r_{i+1},t)]/2$. Density profiles $\rho(r_i,t)$ were then obtained by integrating *Equation (7)* with the forward Euler method using a time step $\Delta t = 20$ min to align with experimental data collection of radial velocity profiles $v_r(r_i,t)$ from *Figure 2B*. The initial conditions were $\rho(r_i,0) = 2700\,\mathrm{cells/mm^2}$ for $r_i < r_{tissue}$ and $\rho(r_i,0) = 0\,\mathrm{cells/mm^2}$ for $r_i > r_{tissue}$, where $r_{tissue}$ is the radius of tissue at the beginning of experiment. For comparison with experimental data (see *Figure 4*), we thresholded the kymographs of simulated density at $100\,\mathrm{cells/mm^2}$, which corresponds to much lower density than a confluent tissue.

### Relating local cell density to vortex centers

For panels (E) and (F) in *Figure 4—figure supplement 1*, we applied a Fourier low-pass filter on vorticity fields, retaining only large-scale vorticity fluctuations (with wavelengths longer than 1 mm). We excluded the tissue edge region (500 µm from the boundary) that is outward polarized and does not exhibit vortical flows. Each point in panels (E) and (F) corresponds to a point in the filtered vorticity field, plotted against the cell density in that point.

### Cell cycle analysis

The Fucci system consists of an RFP and GFP fused to proteins Cdt1 and Geminin, respectively (*Sakaue-Sawano et al., 2008*). Cdt1 levels are high during G1 and low during the rest of the cell cycle, while Geminin levels are high during the S, G2, and M phases (*Sakaue-Sawano et al., 2008*; *Streichan et al., 2014*). After capturing the appropriate fluorescence images, preprocessing was implemented identically for GFP and RFP channels to normalize channel histograms. To determine local cell cycle fraction, we determined the median value of RFP and GFP signal for each cell nucleus and manually selected thresholds for RFP and GFP signals separately to classify cell cycle for each cell as G0-G1-S (RFP above threshold), S-G2-M (RFP below threshold and GFP above threshold), or postmitotic (RFP and GFP below threshold). Local cell cycle fraction of each state could then be easily computed for each PIV pixel. Note that S phase (both RFP and GFP signals above threshold) did not prove to be a reliable feature for segmentation.

### Code and data availability

Data for representative small, large, and ellipse tissues (*Heinrich et al., 2020*) and analysis Matlab scripts (*Heinrich, 2020*) have been made available (copy archived at https://github.com/elifesciences-publications/FreelyExpandingTissues).

## Acknowledgements

DJC acknowledges the National Institutes of Health R35 GM133574-01. RA acknowledges support from the Human Frontiers of Science Program (LT000475/2018-C).

## Additional information

### Funding

| Funder | Grant reference number | Author |
| --- | --- | --- |
| National Institutes of Health | 1 R35GM133574-01 | Daniel J Cohen |
| Human Frontier Science Program | LT000475/2018-C | Ricard Alert |

The funders had no role in study design, data collection and interpretation, or the decision to submit the work for publication.

### Author contributions

Matthew A Heinrich, Data curation, Software, Formal analysis, Validation, Investigation, Visualization, Methodology, Writing - original draft, Writing - review and editing; Ricard Alert, Formal analysis, Validation, Methodology, Writing - original draft, Writing - review and editing; Julienne M LaChance, Data curation, Software, Formal analysis, Investigation, Visualization, Methodology; Tom J Zajdel, Formal analysis, Investigation, Visualization, Methodology, Writing - original draft; Andrej Košmrlj, Formal analysis, Supervision, Funding acquisition, Methodology, Writing - original draft, Project administration, Writing - review and editing; Daniel J Cohen, Conceptualization, Resources, Data curation, Software, Formal analysis, Supervision, Funding acquisition, Validation, Investigation, Visualization, Methodology, Writing - original draft, Project administration, Writing - review and editing

### Author ORCIDs

Matthew A Heinrich (ID) https://orcid.org/0000-0002-9041-5554
Ricard Alert (ID) https://orcid.org/0000-0002-1885-9177
Andrej Košmrlj (ID) https://orcid.org/0000-0001-6137-9200
Daniel J Cohen (ID) https://orcid.org/0000-0001-5819-1135

### Decision letter and Author response

Decision letter https://doi.org/10.7554/eLife.58945.sa1
Author response https://doi.org/10.7554/eLife.58945.sa2

## Additional files

### Supplementary files

- Transparent reporting form

### Data availability

Raw datasets for each figure are available at Zenodo (https://zenodo.org/record/3858845) and can be used to reconstruct our analyses using the code that we also provide. Key analysis code provided at our github repository: https://doi.org/10.5281/zenodo.3861843.

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
