## [Decision Letter]

**Acceptance summary:**

This study investigates how a cellular colony's initial size affects its expansion through cell migration and proliferation, which become decoupled from its original size. This analysis highlights the importance of emerging vortices within the colonies and how anisotropy can affect tissue boundaries within cellular monolayers. While these studies were conducted in flat tissue culture monolayers, they may impact how cellular patterns in tissues may arise.

**Decision letter after peer review:**

Thank you for submitting your article "Size-dependent patterns of cell proliferation and migration in freely-expanding epithelia" for consideration by *eLife*. Your article has been reviewed by three peer reviewers, and the evaluation has been overseen by a Reviewing Editor and Didier Stainier as the Senior Editor. The following individuals involved in review of your submission have agreed to reveal their identity: Alexandre Kabla (Reviewer #3).

The reviewers have discussed the reviews with one another and the Reviewing Editor has drafted this decision to help you prepare a revised submission.

Summary:

The manuscript by Heinrich et al. investigates how a tissue's initial size affects its expansion through cell migration and proliferation, which become decoupled from its original size. This analysis highlights the importance of emerging vortices within the colonies and how anisotropy can affect tissue boundaries within cellular monolayers. Overall, all three reviewers found your paper to be well done and of potential interest to a variety of scientists and felt your approach was insightful and quite thorough.

Revisions:

Together, the reviewers converged on a few points that we felt would not require necessarily any extra experimentation but could be obtained from your current data. These points focus on:

1) Study more systematically the relationship between curvature at the boundary and radial velocity, comparing round and oval shapes. Furthermore, how large does the tissue need to be for the radial velocity to be constant?

2) Directly compare round tissues that have the same current size but originate from different original sizes to determine if differences derive from original size or current size.

3) Please add more statistical analysis, instead of generalised statements.

4) Please add more methods regarding PIV.

5) Please add more discussion regarding the limitations of the model.

While these are the points, we would like you to address, I've not deleted the individual responses, as they may be more precise in addressing how you will do this, given I did not serve as a reviewer, just an editor here.

Reviewer #1:

This manuscript by Heinrich et al. explores how the initial size of a tissue affects its expansion through cell migration and proliferation. In their system, cell dynamics and proliferation at the outer edge of the tissue are decoupled from its original size. However, the initial size of the culture has an effect on how the inner core of the tissue behaves. The manuscript is well written, and the work is interesting and original with potential implications for development. However, I believe that some additional work is needed to make the paper suitable for *eLife*:

1) Although for the most part the data are nicely quantified, for a journal of this category statistical tests should be used to draw conclusions. For example,: “This model fits our data well”. This is qualitative and the authors should provide some objective metric of the goodness of fit. Same concern in subsection “Spatiotemporal dynamics of migration speed and radial velocity” were the authors use qualitative words like “similar” or “practically identical” rather than statistical tests to compare their data. This is a general comment that applies to many other comparisons established in the text, for example those related to Figure 3D in subsection “Vortex kinematics” or the correlation in Figure 3E.

2) In Figure 1—video 3, it seems that the oval tissues grow anisotropically with areas of greater curvature growing slower. I do appreciate that small and large tissues have different curvatures yet the same radial velocity, but what is the radial velocity for the experiments shown in 1E? Is it the same across tissues with different shapes? Is it homogeneous within tissues with variable curvature (higher aspect ratio)? Given that video, could changes in curvature (or other factors) affect the speed of migration and consequently the expansion of those tissues?

3) I think this is briefly mentioned in the Discussion but it is important to directly compare the behaviour of large and small tissues not only over time but also when their sizes are equivalent to demonstrate that any effect arises from the original size of the tissues and not from their current size. There should be a thorough analysis of this in the main Results section.

4) It would be good to back up the statement that vortices nucleate in low-density regions regardless of their location, with some quantifications rather than a couple of representative examples.

Reviewer #2:

In this study, Heinrinch et al. analyzed population dynamics of epithelial cells. They focused on the role of colony size, cell cycle and tissue expansion. Overall, they provide an elegant and interesting experimental study to describe the expansion of cellular monolayers. I could recommend the publication of the paper since it would be of interest for a large community. In particular, the emergence of vortices inside the colonies is very interesting and their results could shed new light on the formation of tissue boundaries within cellular monolayers. The analysis of the vortices is very complete and rigorous, and the cell cycle study is well addressed with clear quantification. I previously reviewed this paper for another journal, and I think that the authors made some efforts to improve their manuscript. In particular, the modeling part provides an interesting new piece of work. I still have a few comments to clarify and strengthen some of the points presented here before publication.

a) The role of cell cycle and expansion should be more discussed. In particular, the role of colony size and cell cycle remains largely unexplained. Experiments using synchronized cells or blocking proliferation could be helpful.

b) Additional experiments using drugs blocking cell division, acto-myosin contractility, actin polymerisation and perturbing cell-cell junctions may be useful to provide a deeper understanding.

c) It would be interesting to provide data on single cell shapes (anisotropy, spreading…) for small and large colonies over space and analyze the potential changes of cellular shapes as a function of their position in the monolayers. Do the authors observe changes in the cell shape anisotropy on small and large tissues? Along this line, the analysis of height of the colonies in the different configurations would be helpful. Do cells exhibit any changes in their apico-basal polarity depending on the size of the tissue and their position that could explain the different behaviors?

d) The sudden decrease in radial velocity could be due to cell division since the duration of cell cycle is about 16 hours in MDCK. It is surprising that this time of inflection is constant for both small and large colonies with varying density. Any explanation? Have the authors tried this for large colonies of varying density as they did for small ones?

e) Could the authors check if the propagation length is constant in both small and large tissues? How does it compare with the correlation length of MDCK cells?

Reviewer #3:

The paper from Heinrich et al. presents a rich analysis of the expansion and proliferation of epithelia. The authors use a commonly studied model system for wound healing and collective migration, MDCK epithelia on a flat substrate, and carefully analyse the behaviour of large (millimetre size) unconfined patches of cells. They map the displacement field of the cells across the tissue as well as monitor the cell cycle. This is to my knowledge the first paper to combine such analysis in a systematic manner at a large scale. They report novel behaviours, such as the emergence of different patterns of behaviours in the rim (radial movements) and in the bulk (rotational movements). More importantly, they clearly demonstrate that size matters in the way such partition of behaviour occurs, and in how this patterning evolves over time. A model provides a physical interpretation of some of these observations, although not all aspects of the work are well explained at this point, as mentioned by the authors themselves. This is a very nice paper which will prompt further discussions (and work) in the community. I recommend its publication.

I have very few concerns about the work. The paper is generally well written and a pleasure to read. Clearly a lot of work has been put into it, and although I would want to know more about certain aspects out of curiosity, it would be unfair to expect this to be included here.

1) The paper is specifically about large "millimetre-scale" epithelia, and I was wondering how large they need to be to match this criterion? When is it too small and what sets this length scale? I would expect that small patches would not be able to expand in size with the same radial velocity as the large populations. Do you have any experimental evidence about this?

2) Another point I would recommend clarifying concerns the calculation of the speed (e.g. on Figure 2). The PIV method section indicates what the spatial resolution is (~100µm by 100µm) but the temporal resolution could be made more explicit. I would expect the results to depend on these length-scales and timescales. How were these windows selected? Could data be included to show that the conclusions are robust with respect to this choice?

3) The modelling presented in the paper does a great job summarising part of the results. However, more could be said about what it fails to reproduce and why.

[Editors' note: further revisions were suggested prior to acceptance, as described below.]

Thank you for resubmitting your work entitled "Size-dependent patterns of cell proliferation and migration in freely-expanding epithelia" for further consideration by *eLife*. Your revised article has been evaluated by Didier Stainier (Senior Editor) and a Reviewing Editor.

The manuscript has been improved but there are some remaining issues that need to be addressed before acceptance, as outlined below:

1) Regarding the relationship between curvature and edge speed, the authors find that the normal velocity is not constant along the ellipse edge, but they use a constant velocity in their model. They should address this briefly in the Results section and explain why they are able to make the assumption of constant velocity in the model. For example, by measuring if the radius of curvature along the ellipses is mainly greater than 1mm (which I imagine it is) and therefore a constant velocity can be assumed as shown in Figure 1—figure supplement 2B.

2) Since a major claim is that the behaviour of the tissues was all due to their original size rather than their current size, it is important to directly compare the behaviour of tissues that were originally large and small at the time point. Is the cell density significantly different for large and small tissues when they are the same size? It is hard to see from the figure. Also, Figure 6 is entirely addressed in the Discussion section rather than in the results, despite its importance. Why is it not in the Results section?

---

## [Author Response]

Revisions:Together, the reviewers converged on a few points that we felt would not require necessarily any extra experimentation but could be obtained from your current data. These points focus on:1) Study more systematically the relationship between curvature at the boundary and radial velocity, comparing round and oval shapes. Furthermore, how large does the tissue need to be for the radial velocity to be constant?

We performed new analyses specifically relating curvature to normal expansion velocity including our elliptical tissue data. These data are seen in Figure 1—figure supplement 2. They show normal velocity as a function of curvature and demonstrate that the normal velocity is independent of curvature except in the extreme curvature cases on the highest aspect ratio ellipses. A fuller description can be found in paragraph four of the Results section, and we also provide further context in paragraph two of the Discussion section. With regards to the minimum tissue size needed for constant radial velocity, our analyses suggest tissue diameters of at least 1 mm given that we calculate the stable boundary zone is ~500 µm. This distinction has been added to the Discussion section.

2) Directly compare round tissues that have the same current size but originate from different original sizes to determine if differences derive from original size or current size.

This is an important consideration and insightful question that frequently comes up during presentations of this work, so we devoted an additional figure (Figure 6) to more quantitatively explore this. Our new analyses specifically explore the question of tissues of similar current size but different initial sizes with respect to migration speed, vortex power, cell density, and cell cycle (Figure 6). These new analyses further highlight the importance of history and context in understanding tissue behaviors and make it clear that initial tissue size and mechanical history, rather than current size, drives tissue behavior. A fuller discussion can be found in paragraph three of the Discussion section.

3) Please add more statistical analysis, instead of generalised statements.

Proper statistics are critical, so we have conducted new statistical analyses for all comparative studies. No prior claims have changed, but we now have more quantitative standards to support them. Specific analyses are as follows. Model goodness-of-fit is now evaluated using Chi-squared analysis, which can be seen in Figure 1E, and interpreted as Chi-squared values < 1 representing reasonable fits. Comparison of large and small tissue speeds and radial velocities were assessed by comparing the difference between the datasets compared to their respective standard deviations, as seen in Figure 2D. T-testing via the Mann-Whitney test was performed to compare distributions for vorticity (Figure 3D) and highlight the strongly statistically significant difference between them (*p* < 0.0001). Further analysis of the spatial correlation between density and vorticity resulted in a new panel (Figure 4—figure supplement 1E, F) displaying the relationships more quantitatively and across the whole dataset. All analyses are discussed in the Materials and methods section.

4) Please add more methods regarding PIV.

We appreciated this common as selection and assessment of PIV parameters is non-trivial and deserves more detailed discussion to aid in reproducibility. To address this, we carefully evaluated the effects of PIV parameters on our results and verified that changing the interrogation box did not appreciably alter the large-scale features or structures of the flow fields. That the larger PIV window is sufficient is especially useful because it can dramatically reduce computation time for large times relative to a smaller window size. These data are now presented in Figure 2—figure supplement 1, and the approach is described in the Materials and methods section.

5) Please add more discussion regarding the limitations of the model.

We discussed the limitations of the model in paragraph four of the Discussion section. We now explain that our model does not account for the cell density field, whereas our data show that vortex formation co-occurs with low-density regions. Again, our model is not intended to be exhaustive, but to capture key mechanistic details with as few parameters as possible. Our data and simulation results therefore call for the development of more detailed models to capture the relationship between vortex formation and cell density.

[Editors' note: further revisions were suggested prior to acceptance, as described below.]

The manuscript has been improved but there are some remaining issues that need to be addressed before acceptance, as outlined below:1) Regarding the relationship between curvature and edge speed, the authors find that the normal velocity is not constant along the ellipse edge, but they use a constant velocity in their model. They should address this briefly in the Results section and explain why they are able to make the assumption of constant velocity in the model. For example, by measuring if the radius of curvature along the ellipses is mainly greater than 1mm (which I imagine it is) and therefore a constant velocity can be assumed as shown in Figure 1—figure supplement 2B.

This is important to clarify, so we have the following to address it. As noted, high curvature only applies to a small region of the 1:8 ellipses, which will blunt over time as it grows, so our constant velocity model still fits the overall area expansion data well.

“Such high curvatures are concentrated around the major axes of our elliptical tissues. However, most of the tissue edge has a smaller curvature, and therefore advances at a curvature-independent speed. Further, even high curvature regions blunt due to expansion over time (see Figure 1—video 3). As a result, our model with a single edge speed 𝑣_𝑛_ ≃ 29.5 𝜇m/h is sufficient to capture the area expansion of both circular and elliptical tissues (Figure 1E).”

2) Since a major claim is that the behaviour of the tissues was all due to their original size rather than their current size, it is important to directly compare the behaviour of tissues that were originally large and small at the time point. Is the cell density significantly different for large and small tissues when they are the same size? It is hard to see from the figure.

It is true that the difference in density is less apparent in Figure 6C, and we have added additional discussion specifically exploring this point and emphasizing both the rate of change of density and the fact that the primary density phenotypes occur in the early stages of growth where smaller tissues experience a drop in density while larger tissues experience a monotonic increase. These are summarized below.

“…at equal current sizes, while absolute cell densities in the tissue centers share some overlap, it is notable that the rate of density change at the tissue center is increasing faster in initially-small tissues than in initially-large tissues (Figure 6C). However, the most striking differences in cell density evolution occur not at equal current sizes but during the early stages of tissue expansion: whereas the cell density at the center of large tissues increases at all times, the center of small tissues features a marked density decrease between ∼ 8 and ∼ 24 h (Figure 4A,B).”

Also, Figure 6 is entirely addressed in the Discussion section rather than in the results, despite its importance. Why is it not in the Results section?

We felt strongly that Figure 6 belongs in the Discussion section because it represents a compilation and comparison of data already presented in the prior figures in different forms. In other words, Figure 6 does not show new measurements but rather reanalyzes the data in all the previous figures in a way that showcases history effects. Thus, it is a figure that very much engages with, and emerges from, the points raised in the Discussion section relating to history effects, putting our results into a broader context. Therefore, we feel that moving Figure 6 to the Results section would weaken its impact and the Discussion section overall. That said, we can move it to the Results if necessary.